# *Trichomonas gallinae* Kills Host Cells Using Trogocytosis

**DOI:** 10.3390/pathogens12081008

**Published:** 2023-08-02

**Authors:** Chen Xiang, Yi Li, Shengfan Jing, Shuyi Han, Hongxuan He

**Affiliations:** 1National Research Center for Wildlife-Borne Diseases, Institute of Zoology, Chinese Academy of Sciences, Beijing 100101, China; xiangchen@ioz.ac.cn (C.X.); hanshuyi@ioz.ac.cn (S.H.); 2University of Chinese Academy of Sciences, Beijing 100101, China; 3College of Veterinary Medicine, Hebei Agricultural University, Baoding 071000, China; ly_gogzuo@163.com (Y.L.); 19948169716@163.com (S.J.)

**Keywords:** *Trichomonas gallinae*, trogocytosis, PI3K inhibitor, cysteine protease inhibitor

## Abstract

*Trichomonas gallinae* (*T. gallinae*) is an infectious parasite that is prevalent worldwide in poultry and can cause death in both poultry and wild birds. Although studies have shown that *T. gallinae* damages host cells through direct contact, the mechanism is still unclear. In this study, we found that *T. gallinae* can kill host cells by ingesting fragments of the host cells, that is, by trogocytosis. Moreover, we found that the PI3K inhibitor wortmannin and the cysteine protease inhibitor E-64D prevented *T. gallinae* from destroying host cells. To the best of our knowledge, our study has demonstrated for the first time that *T. gallinae* uses trogocytosis to kill host cells. Understanding this mechanism is crucial for the prevention and control of avian trichomoniasis and will contribute to the development of vaccines and drugs for the prevention and control of avian trichomoniasis.

## 1. Introduction

The flagellated protozoan parasite *Trichomonas gallinae* (*T. gallinae*) is the causative agent of avian trichomoniasis, a disease that poses a significant threat to both wild birds and poultry [1,2,3]. This parasite is predominantly found in Columbiformes [2], Passeriformes [2,4], and raptors [5,6]. Transmission of *T. gallinae* primarily occurs through the ingestion of contaminated food and water, with raptors often contracting the parasite by preying on infected birds [7]. Early symptoms of the disease include the formation of caseous masses and ulcers in the oropharyngeal cavity, crop, and proximal esophagus. These symptoms can lead to starvation in birds, as the resulting tissue inflammation and necrotic masses hinder their ability to swallow food [8,9]. A characteristic pale yellow, cheese-like change in the oropharynx serves as the primary diagnostic indicator of avian trichomoniasis [10].

*T. gallinae* has spread worldwide, with reported cases in Iran [9], Hungary, Romania [11], Africa [12], and China [13]. Mortality has also been widely reported [14]. The main treatment for avian trichomoniasis involves nitroimidazole drugs [15], but long-term administration can lead to drug resistance and residues. Unfortunately, no vaccine is currently available for the treatment and prevention of *T. gallinae*. Therefore, gaining a comprehensive understanding of the pathogenic mechanisms of *T. gallinae* in the host is crucial.

Recently, a novel mechanism of host cell damage known as trogocytosis has been discovered in several parasites, including *Entamoeba histolytica* [16], *Plasmodium falciparum* [17], and *Toxoplasma gondii* [18]. The term “trog” in trogocytosis, derived from the Greek word for “to nibble,” suggests a block-by-block uptake mechanism. Consequently, this phenomenon has garnered escalating interest across various biological research domains, encompassing immunology, microbiology, neurology, and developmental biology [19,20,21,22,23,24,25]. Trogocytosis, possibly a form of endocytosis, involves the rapid uptake of a fragment of one cell by another, relying on cell-to-cell contact [26]. As of now, trogocytosis has not been reported in Trichomonas; only nibbing has been observed in *Trichomonas vaginalis* [27].

Currently, the primary treatment for avian trichomoniasis involves the use of nitroimidazole drugs [15]. However, prolonged usage can result in drug resistance and the presence of drug residues. To date, no vaccine has been developed for the treatment and prevention of *T. gallinae*. Therefore, gaining a comprehensive understanding of the pathogenic mechanism of *T. gallinae* holds great significance. The present study demonstrates that *T. gallinae* kills host cells through a direct contact-dependent, trogocytosis-mediated mechanism instead of phagocytosis. *T. gallinae* gnaws at host cells during this process, regardless of whether they are alive or dead. Additionally, E-64D, an inhibitor of cysteine proteases, has demonstrated substantial inhibitory effects on trogocytosis while exhibiting low toxicity to cells. Therefore, it could be a promising candidate for the development of a therapeutic agent against *T. gallinae*. In conclusion, these findings unveil a novel mechanism by which *T. gallinae* induces host cell death through trogocytosis.

## 2. Materials and Methods

### 2.1. Cell and Parasite Culture

The cell line used in this study was the Chicken Liver Hepatocellular Carcinoma cell line (LMH), obtained from the Beijing BeNa Culture Collection in China. LMH cells were cultured in Dulbecco’s Modified Eagle Medium (Thomas Scientific, Swedesboro, NJ, USA) supplemented with 10% fetal bovine serum (FBS) and 1% penicillin/streptomycin [28]. The cell culture incubator was maintained at 37 °C with 5% CO_2_. The *T. gallinae* strain used in the study was kept in our laboratory and originated from a pigeon. Tryptose-Yeast-Maltose (TYM) medium supplemented with 10% FBS and 1% Penicillin/Streptomycin was used to cultivate *T. gallinae*, and the culture was incubated at 37 °C [29]. *T. gallinae* was passaged every two days and maintained at a density of approximately 1 × 10^5^ cells/mL.

### 2.2. LMH-Trichomonas gallinae Co-Culture Conditions

The same medium and conditions employed for LMH cell culture were utilized for the co-culture of *T. gallinae* trophozoites with LMHs. After the LMH cells had reached full confluence, the medium was aspirated, and 0.25% trypsin was added to the cells for one minute of digestion. Subsequently, the cells were evenly distributed into 24-well plates, with approximately 2.5 × 10^5^ cells per well. *T. gallinae* trophozoites were suspended and centrifuged at 1800 rpm for 5 min at low speed. The trophozoites were then resuspended in DMEM complete medium and added to the 24-well plates containing LMHs at a ratio of 1:1, with approximately 2.5 × 10^5^ parasites per well. The co-culture was maintained in a 5% CO_2_ incubator at 37 °C.

### 2.3. Trans-Well Assay

LMH cells were labeled with 20 μM CellTraceTM Red CMTPX (Thermo Fisher Scientific, Waltham, MA, USA) at a 1:100 ratio following the manufacturer’s instructions. The labeled LMHs were then inoculated into 500 μL 24-well plates at a density of 2.5 × 10^5^ cells per well. *T. gallinae* was labeled with 20 μM 5-chloromethylfluorescein diacetate (Thomas Scientific, Swedesboro, NJ, USA) for 45 min, followed by incubation in complete TYM medium for 30 min. After washing with phosphate-buffered saline (PBS), the parasites were resuspended in DMEM medium. The cells were then seeded into 24-well plates at a density of 2.5 × 10^5^ cells per well. *T. gallinae* and LMH cells were co-incubated in a trans-well assay at a 1:1 ratio. The trans-well device consists of 24-well plates with a 3.0 μm pore size and a 10 μm thick polycarbonate membrane insert (Corning, New York, NY, USA). In the trans-well assay, *T. gallinae* was incubated in the upper well, and LMH cells were incubated in the lower well. Alternatively, *T. gallinae* and a few LMH cells were co-incubated in the upper well, while LMH cells were incubated in the lower well. The lower wells were harvested after a 12 h incubation period at 37 °C. A one-way ANOVA was conducted to assess the statistical significance among the conditions of interest.

### 2.4. Trichomonas gallinae Cytotoxicity Assay

A flow cytometry-based cytotoxicity assay was employed. LMH cells were briefly labeled with 20 μM CellTrace^TM^ Red CMTPX tracer and seeded into 24-well culture plates at a density of 2.5 × 10^5^ cells per well. *T. gallinae* was labeled with a 20 μM CMFDA tracer and incubated in complete TYM medium for 30 min. Subsequently, the ratio of *T. gallinae* and LMH cells in co-culture was adjusted to achieve multiplicities of infection (MOIs) of 0.1, 0.2, 0.5, 1, and 2, and the impact of *T. gallinae* on LMH cell damage at various concentrations was evaluated. The co-cultures were incubated in a CO_2_ incubator at 37 °C for 3, 6, 12, and 24 h. The impact of *T. gallinae* on LMH cell damage at various co-culture times was examined. The ratio of *T. gallinae* to LMH cells was 1:1.

*T. gallinae* was treated with the PI3K inhibitor wortmannin at a concentration of 30 nM for 1 h, while an equal volume of DMSO (Gaylord Chemical, Louisiana, AL, USA) solution was used as a control at the same conditions. Subsequently, the plates were washed twice with PBS and then added to 24-well plates containing LMH cells, with 2.5 × 10^5^ *T. gallinae* per well. The data were obtained using flow cytometry at a MOI of 1. *T. gallinae* was also treated with the cysteine protease inhibitor E-64D at a concentration of 30 nM for 1 h, using the same method as employed for wortmannin treatment.

### 2.5. Determination of Trichomonas gallinae Viability

LMH cell viability was assessed using the Cell Counting Kit-8 (Vazyme, Nanjing, China). LMH cells were added to 96-well plates at a density of 10^5^ cells/mL, and the plates were then incubated at 37 °C for 24 h. A total of 30 nM wortmannin or 30 nM E-64D was added to the 96-well plates at varying concentrations, followed by incubation at 37 °C for 1 h. Subsequently, 10 μL of the CCK-8 solution was added to each well, and the plates were incubated for 2 h at 37 °C in the incubator. Finally, the absorbance at 450 nm was measured using a microplate reader. LMH cell survival was calculated by subtracting the optical density (OD) of each test well from the background OD (blank group), and the OD of each replicate well was then averaged ± standard deviation (SD). Cell survival (%) was calculated as (OD of drugged cells/OD of control cells) × 100.

### 2.6. Flow Cytometry

Brooks et al. demonstrated the suitability of *T. vaginalis*—host cell co-cultures for flow cytometry analysis [30]. As both *T. gallinae* and *T. vaginalis* belong to the genus Trichomonas, we employed flow cytometry to examine their co-cultures with LMHs. Differential dye staining was applied to distinguish between *T. gallinae* and LMHs. LMH cells were labeled with 20 μM CellTrace^TM^ Red CMTPX and inoculated into 24-well plates at a density of 2.5 × 10^5^ cells. Subsequently, *T. gallinae* was labeled with 20 μM CMFDA and added to the 24-well plates containing LMHs at the specified MOI. The co-cultures were incubated for 12 h in a 37 °C incubator. The supernatant and bottom sediment were collected, washed once with PBS, resuspended in serum-free DMEM, and subsequently analyzed on a BD Fortessa. Each sample was mixed with 50 μL of absolute counting microbeads (Invitrogen^TM^), and cell counts were normalized to 10,000 events before analysis. The percentage of LMH killing was calculated using the formula: [(number of live LMH cells in LMH alone conditions—number of live LMH cells in co-culture conditions)/number of live LMH cells in LMH alone conditions] × 100. CMFDA and CellTrace^TM^ Red CMTPX are live cell tracers that fluoresce only in live cells, while dead cells do not exhibit fluorescence in the dot plots. To determine the statistical significance between the relevant conditions, a one-way ANOVA was conducted.

### 2.7. Live Confocal Imaging

Prior to confocal visualization, LMHs or *T. gallinae* were washed with PBS and labeled using fluorescent dyes. We observed whether CMFDA-positive *T. gallinae* contained LMH cell fragments labeled with CellTrace^TM^ Red CMTPX and investigated the process by which *T. gallinae* acquired LMH cell fragments. LMHs were incubated at 42 °C for 1 h to inactivate them and then labeled with 50 µM PI for 20 min. After two rounds of washing, 2–3 drops of anti-fluorescence quencher were added. Prior to imaging, *T. gallinae* was added to LMHs. For live imaging experiments, LMH cells labeled with CellTrace^TM^ Red CMTPX were diluted into single cells and photographed in a bright field for 30 min immediately after the addition of *T. gallinae*. For heat-inactivated LMHs, they were co-cultured with *T. gallinae* for 3 h before being photographed. The imaging of cells was conducted in 35 mm glass-bottom plates containing 100 µL of DMEM medium. The confocal and live imaging were performed using a Zeiss LSM980 inverted microscope equipped with a 63× compound achromatic oil objective and a heated stage. The data were analyzed using Zeiss ZEN software (Carl Zeiss AG, Oberkochen, Germany) (https://www.zeiss.com/microscopy/en/products/software/zeiss-zen.html) (accessed on 2 June 2023).

Three-dimensional live cell imaging of LMH and *T. gallinae* was performed after labeling them with CellTrace^TM^ Red CMTPX and CMFDA, respectively. The X, Y, and Z axes were configured with one layer interval of 0.5 μm, while the *Z*-axis scanning time was not specified due to its extended duration, resulting in no time interval being set. The 3D imaging was conducted on an OLYMPUS FV3000RS microscope equipped with a 60× silicone oil objective, and the obtained results were analyzed using Imaris software (Bitplane, Zürich, Switzerland) (https://www.waisman.wisc.edu/cellular-and-molecular-neuroscience/imaris/) (accessed on 2 June 2023).

### 2.8. Statistical Analysis

The data were analyzed and correlated using GraphPad Prism 9. Differences were considered significant if *p* < 0.05.

## 3. Results

### 3.1. Trichomonas gallinae Damages Host Cells in a Time- and Concentration-Dependent Manner

For many years, *T. gallinae* has been the primary suspect as the sole cause of avian trichomoniasis. However, the mechanism by which *T. gallinae* kills host cells remains unclear [31]. Therefore, we investigated the ability of *T. gallinae* to kill LMHs using an in vitro cytotoxicity assay. Microscopy was employed to examine the interactions of *T. gallinae* with LMHs and oral epithelial cells in pigeons. *T. gallinae* was observed to aggregate around the cells (Figure 1A). Subsequently, *T. gallinae* was labeled with CMFDA, and LMHs were labeled with CellTrace^TM^ Red CMTPX before being co-cultured for 12 h. The results were analyzed using flow cytometry. Consequently, the two-dimensional dot plot revealed the presence of dual fluorescence (CMTPX+CMFDA+, Figure 1B), suggesting the interaction between *T. gallinae* and LMHs and the potential presence of LMHs within *T. gallinae*. Subsequently, the number of surviving LMHs (CMFDA−CMTPX+ cells) was quantified, confirming the cytotoxic effect of *T. gallinae* on LMHs. The number of surviving CMFDA−CMTPX+ cells was determined by comparing the ratio of CMFDA−CMTPX+ cells in LMH cell-independent conditions to those in the presence of *T. gallinae*. A higher percentage of cell death (approximately 90%) was observed (Figure 1C). LMHs were able to survive more successfully by reducing the ratio of MOI (lower *T. gallinae* concentration) (Figure 1D). Additionally, shortening the duration of infection may increase the survival of LMHs (Figure 1C). These results indicate that *T. gallinae* induces damage to LMHs, resulting in their subsequent deaths.

### 3.2. Trichomonas gallinae Killing of LMH Is Contact-Dependent

A previous study by Amin et al. demonstrated that *T. gallinae* inflicts damage to LMHs through direct interactions with the parasites [32]. In this study, we further validated the interaction between *T. gallinae* and LMHs using a trans-well device. The upper and lower cell chambers are separated by a membrane with a 3.0 μm pore size, allowing for continuous medium exchange (Figure 2A, ID: SSYRAdea6e). Our observations revealed a significant reduction in LMH cell death caused by *T. gallinae* when the parasites were placed in a separate chamber (Figure 2B). This led us to speculate whether *T. gallinae* could activate the secretion of specific factors only upon direct contact with LMHs. To investigate this, we co-cultured a small number of LMHs in the lower chamber while *T. gallinae* was placed in the upper chamber (Figure 2B). Compared to co-culturing, we observed a notable decrease in the number of LMH deaths. However, the number of LMHs that died was still significantly higher than those cultured alone, suggesting that *T. gallinae*’s secreted factors can also cause cell death in a non-contact-dependent manner. These findings highlight the importance of exposure-related cell killing in *T. gallinae*’s pathogenicity.

### 3.3. Trichomonas gallinae Kills LMH Cells by Swarming and Trogocytosis

While LMHs were slightly larger than *T. gallinae*, the CMTPX+CMFDA+ double-positive population exhibited a smear gradient instead of a clear population shift (Figure 1B), which contradicted the notion of *T. gallinae* engulfing whole cells. Instead, these findings support a recently described parasitic strategy known as trogocytosis (trog = gnawing), utilized by *T. gallinae* to kill host cells [21,33,34]. To investigate whether *T. gallinae* employed trogocytosis, a cell-killing mechanism not previously attributed to the parasite, we employed confocal microscopy to observe CMTPX+CMFDA+ events. In comparison to whole LMH cells, we observed that CMTPX+CMFDA+ cells are fragments of *T. gallinae* that seem to contain LMHs (Figure 3A). Moreover, we examined the interaction between live and heat-inactivated (dead) LMHs with *T. gallinae* to explore any differences between the two conditions. Interestingly, we found no significant morphological distinctions between live and heat-inactivated cells in the presence of *T. gallinae* (Figure 3A,B). Similarly, in the case of *T. gallinae* co-cultured with heat-inactivated cells, we observed small patches of debris from LMHs (Figure 3B). These results contrast with the phagocytosis of dead cells by *E. histolytica*.

A real-time imaging strategy was employed to monitor the interactions between *T. gallinae* and LMHs, aiming to ascertain whether fragments of LMHs would be present within *T. gallinae* prior to LMH cell death. Bright-field imaging was used to observe *T. gallinae* labeled with CMFDA and LMHs labeled with CellTrace^TM^ Red CMTPX. Interestingly, we observed LMHs being surrounded by free-swimming parasites (Figure 4A,B) and noted that red signals were transmitted to *T. gallinae* and accumulated there (Figure 4A,B). These findings are inconsistent with the interpretation that the observed spots of living LMH material in *T. gallinae* are degraded whole cells after phagocytosis, suggesting that the interaction involves “biting” before the death of LMHs. Furthermore, our results provide evidence that *T. gallinae* employs trogocytosis rather than phagocytosis to kill host cells.

### 3.4. Cysteine Protease and PI3K Mediate T. gallinae Trogocytosis

According to the data presented in Figure 2B, *T. gallinae* primarily eliminates LMHs through a contact-dependent mechanism, as evident by the clear shift of CMTPX−CMFDA+ in *T. gallinae* to the CMTPX+CMFDA+ quadrant. This indicates a possible trogocytosis process where CMTPX+ LMHs are engulfed by CMFDA+ *T. gallinae*, as depicted in Figure 4A. To validate this observation, we employed inhibitors targeting phosphoinositide 3-kinase (PI3K) and cysteine proteases (CPs), both of which play crucial roles in trogocytosis [16,35]. Consistent with our expectations, treatments with wortmannin and E-64D substantially decreased the mortality of LMHs caused by *T. gallinae*. Conversely, in the control group of *T. gallinae*, cell mortality remained significantly higher. Notably, the parasites treated with inhibitors demonstrated significantly reduced feeding activity, as shown in Figure 5A,C. We hypothesized the potential therapeutic application of wortmannin against *T. gallinae*. However, safety tests on cells revealed that wortmannin exhibited greater cytotoxicity compared to DMSO, as depicted in Figure 5B. Furthermore, we investigated the impact of pre-treating *T. gallinae* with the CPs inhibitor E-64D on its ability to interact with LMH. Utilizing flow cytometry, we observed a significant reduction in the frequency of CMTPX+CMFDA+ double-positive events with E-64D treatment, as depicted in Figure 5C. Similarly, the safety of E-64D was assessed, and no considerable cellular damage was observed (Figure 5D). Collectively, these findings strongly suggest that *T. gallinae* induces damage to host cells through the process of trogocytosis.

## 4. Discussion

Despite the high prevalence of *T. gallinae* infection, the pathogenic mechanism of this parasite remains poorly understood. While trogocytosis has been reported in various parasites and is known to play a key role in damaging host cells [19], it has not been studied in the context of *T. gallinae* until now. In this study, we demonstrate that it kills host cells by employing a cooperative swarming behavior around the host cells, followed by attachment and trogocytosis. This process leads to increased mortality of host cells when a large number of *T. gallinae* are present. Furthermore, we found that this cell-killing mechanism is mediated by *T. gallinae* through PI3K and cysteine proteases. Taken together, our findings strongly suggest that *T. gallinae* acquires host cells via trogocytosis.

Our findings demonstrate that *T. gallinae* inflicts damage on host cells through direct contact (Figure 2B). This observation aligns with previous reports of direct contact damage to LMHs by *T. gallinae* [36]. The oral mucosa serves as the host’s primary innate defense against *T. gallinae*, and the initiation of a symptomatic intraoral infection requires the parasite’s penetration of the salivary mucus layer. Adhesion to mucous membranes represents a critical step in *T. gallinae* invasion through contact-dependent pathogenic mechanisms. Notably, *T. gallinae* expresses several cysteine proteases (CPs), which are known to play essential roles in various biological functions, including host cell adhesion, pathogenicity, and virulence [37]. Our results also indicate that CPs are released only upon contact of *T. gallinae* with host cells, as demonstrated by the minimal killing of host cells in the absence of contact (Figure 2B). Additionally, treatment of *T. gallinae* with the CPs inhibitor E-64D significantly reduced the mortality rate of host cells compared to untreated parasites (Figure 5B). This finding is consistent with previous reports suggesting the involvement of CPs in the trogocytosis process of *E. histolytica* in lysogenic tissues [35].

Trogocytosis, a well-known cell-to-cell contact mechanism in the pathogenesis of protozoan parasites, involves a dynamic process dependent on actin polymerization, which can be impeded by inhibiting PI3K [38]. The activation of PI3K is directly associated with cell-cell adhesion, leading to the regulation of F-actin redistribution and assembly. Consequently, its inhibitory effect primarily affects F-actin at the cell-cell contact site, resulting in reduced trogocytosis due to destabilized cell-cell interactions [38]. Our findings provide evidence that *T. gallinae* kills host cells through trogocytosis. Moreover, we observed that the use of the PI3K inhibitor wortmannin diminished *T. gallinae*’s ability to induce host cell death. These results strongly suggest a role for PI3K in *T. gallinae*-induced injury (Figure 5A), corroborating prior research findings [38].

The current pharmacological treatment of *T. gallinae* mainly relies on metronidazole and other nitroimidazoles. However, their usage as single-type drugs can lead to drug resistance with prolonged administration [39]. In our research, we employed two inhibitors, wortmannin and E-64D, to impede *T. gallinae* from damaging host cells by interfering with trogocytosis. Notably, there have been no reports on the potential use of wortmannin or E-64D as therapeutic agents in prior research. Assessing the safety of both inhibitors, we found that E-64D had a low toxic effect on cells (Figure 5D). E-64D, a widespread cysteine protease inhibitor derived from a natural fungal product, has been investigated for myotonic dystrophy treatment. However, it has not demonstrated success in human clinical trials, despite ongoing research for its potential application in spinal cord injury, stroke, and Alzheimer’s disease [40]. Given its history of use in treating several diseases, we speculated on its potential as a drug for avian trichomoniasis. Further pharmacological and pharmacokinetic studies on E-64D will be pursued. However, due to the significant cellular lethality associated with wortmannin, it is not considered a viable treatment for avian trichomoniasis.

In conclusion, the assay successfully demonstrated that *T. gallinae* aggregates inflict damage on host cells, undergo trogocytosis, and ultimately lead to host cell death. This process involves the participation of CPs and PI3K. E-64D, a broad-spectrum CPs inhibitor, holds promise as a potential drug for future *T. gallinae* treatment. Subsequent research is required to elucidate the specific mechanism of CPs in *T. gallinae* trogocytosis. Investigating the in vitro interactions between host cells and *T. gallinae* stimulated by cytokines may pave the way for innovative strategies to impede *T. gallinae* invasion. Moreover, this mechanistic understanding could significantly contribute to vaccine development and the enhancement of therapeutic approaches. The simultaneous reduction of *T. gallinae* infection and its complications in wild birds and poultry, along with the improvement of vaccine efficacy and treatments, can be achieved through these advancements.

## 5. Conclusions

Our primary finding is that *T. gallinae* inflicts damage on host cells through trogocytosis. Additionally, we discovered that the CPs inhibitor E-64D exhibits a high safety profile on host cells, indicating its potential as a drug candidate for avian trichomoniasis treatment.

## Figures and Tables

**Figure 1 pathogens-12-01008-f001:**
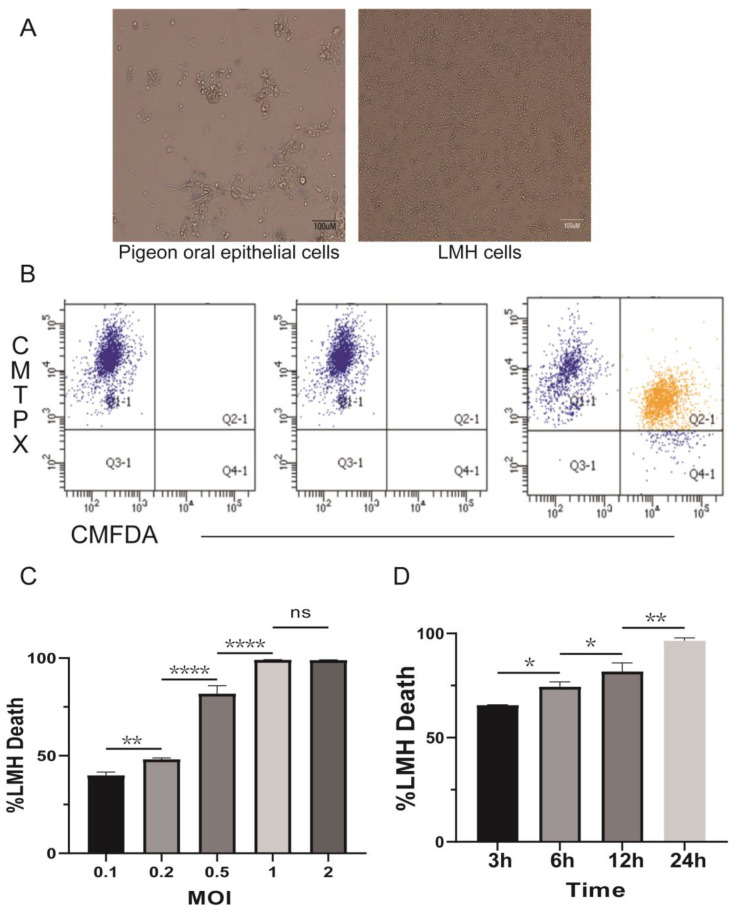
*T. gallinae* induces host cell death. (**A**) Interaction of *T. gallinae* with host cells observed under optical microscopy. (**B**) A representative flow cytometry plot showing the results of the cytotoxicity assay using an MOI of 1. Surviving LMHs were identified as CMTPX+CMFDA−. (**C**,**D**) Percent cytotoxicity calculated based on the number of surviving LMHs from the plot in (**B**). (**C**) The survival rate of LMHs was assessed at different MOIs, which represent the ratio of parasites to host cells. (**D**) Parasites and LMHs were co-cultured at different times to assess the survival of LMHs. All data are presented as mean ± SD. ns: no significance, * *p* < 0.05, ** *p* < 0.01, and **** *p* < 0.01.

**Figure 2 pathogens-12-01008-f002:**
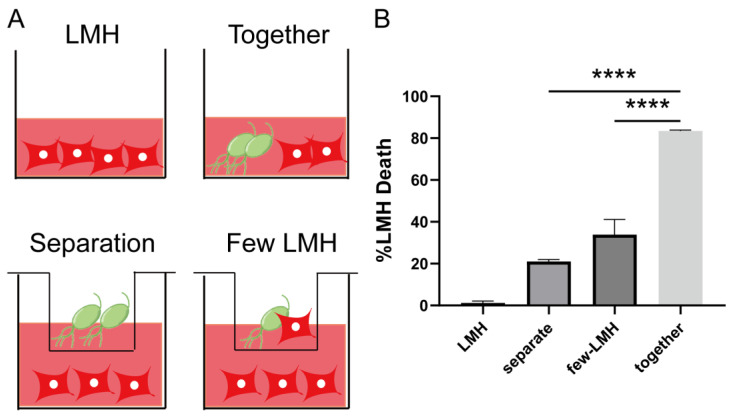
The killing of LMHs by *T. gallinae* is shown to be contact-dependent. (**A**) *T. gallinae* and LMHs were labeled with CMFDA or CellTrace^TM^ Red CMTPX, respectively. They were then co-incubated for 12 h in trans-well plates at the indicated MOI, together in the bottom chamber or separately in the bottom chamber, with *T. gallinae* in the top chamber and LMHs in the bottom chamber. The cultures were incubated at the equivalent MOI. (**B**) The survival rate of the LMHs listed in A was calculated. Statistical significance was denoted as follows: **** *p* < 0.001.

**Figure 3 pathogens-12-01008-f003:**
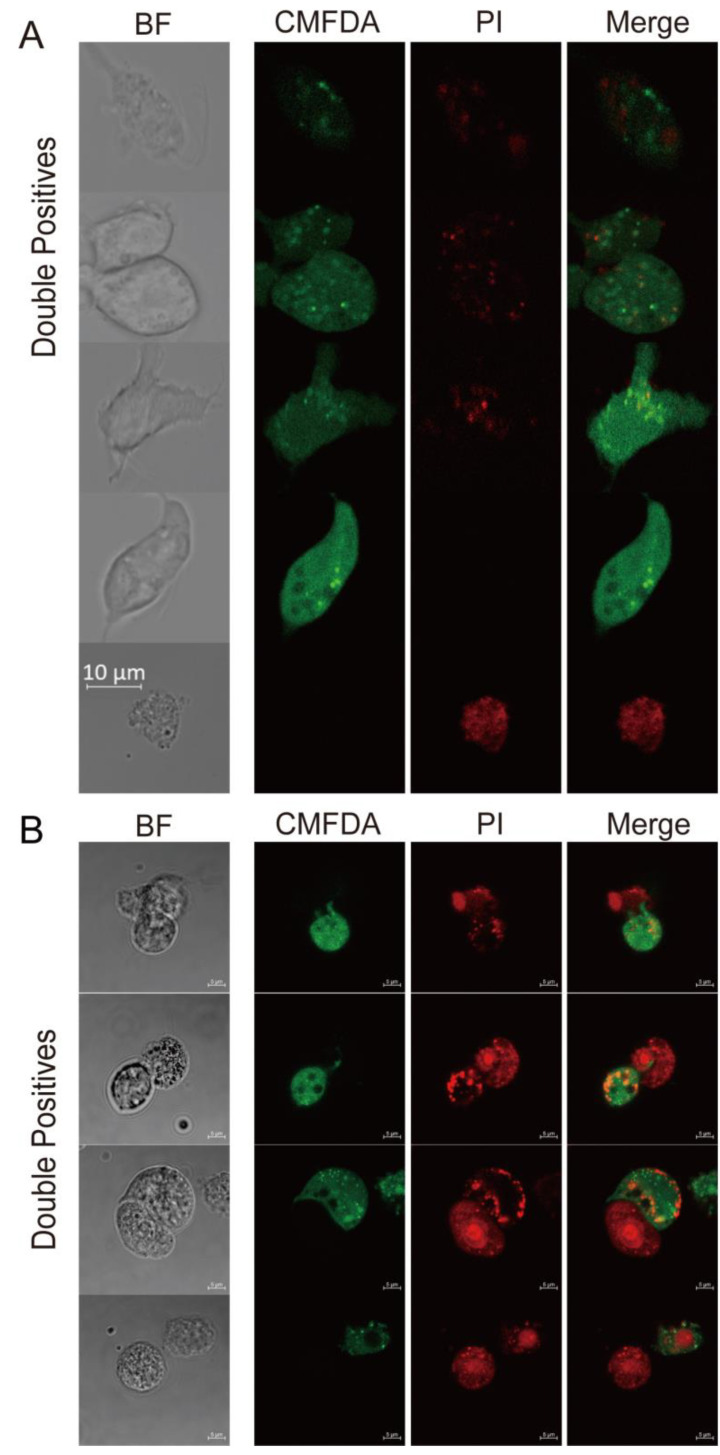
*T. gallinae* internalized fragments of live/dead LMH cells. Co-cultured under CMFDA or CMTPX/PI labeling for 3 h, *T. gallinae* and LMHs were observed using bright field imaging (BF). (**A**) Representative images depict populations of CMTPX+CMFDA+ (double positive), CMTPX−CMFDA+ (*T. gallinae*), and CMTPX+CMFDA− (LMH). (**B**) Additional representative images display PI+CMFDA+ (double positive), PI−CMFDA+ (*T. gallinae*), and PI+CMFDA− (LMH) populations. The interactions between the two were captured at selected time points using 63× magnified 2D field videos. Scale bar = 10 μm.

**Figure 4 pathogens-12-01008-f004:**
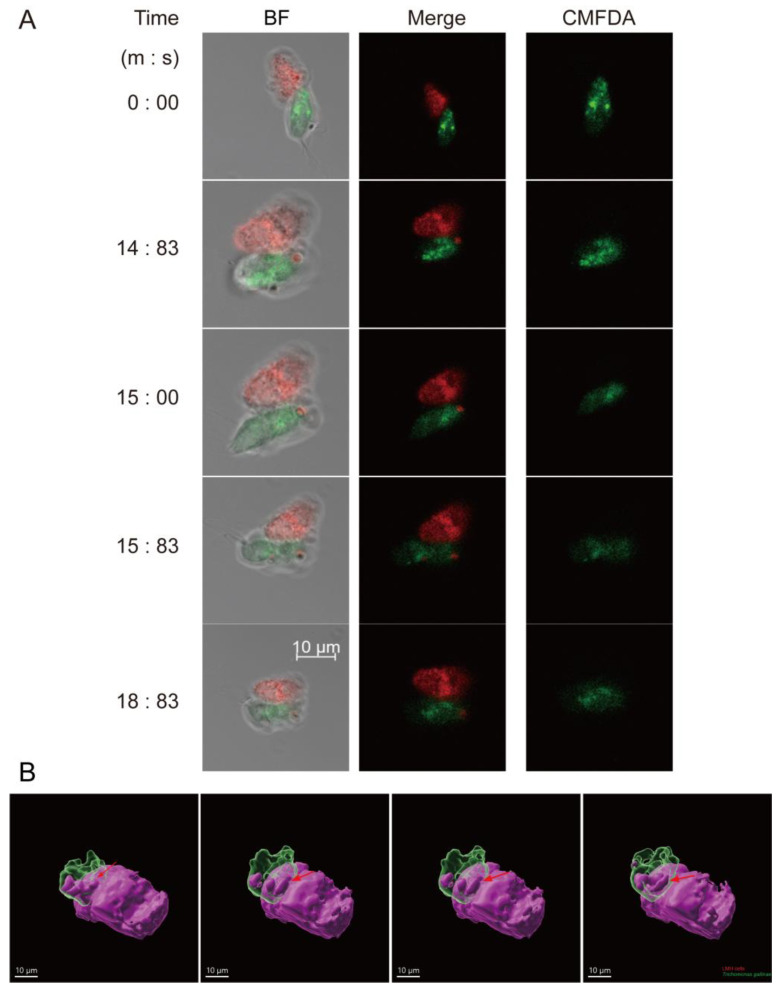
Depicts the interaction process between *T. gallinae* and LMH cells. (**A**) LMHs labeled with CMTPX and CMFDA are internalized by *T. gallinae* in the form of fragments, as observed in selected time points of 2D live video at 63× magnification (Figure 1A). (**B**) Three-dimensional reconstructions of CMTPX-labeled LMHs and CMFDA-labeled *T. gallinae* showcase ingested fragments (arrows) (Figure 1B). The scale bar represents 10 µm. Scale bar = 10 um.

**Figure 5 pathogens-12-01008-f005:**
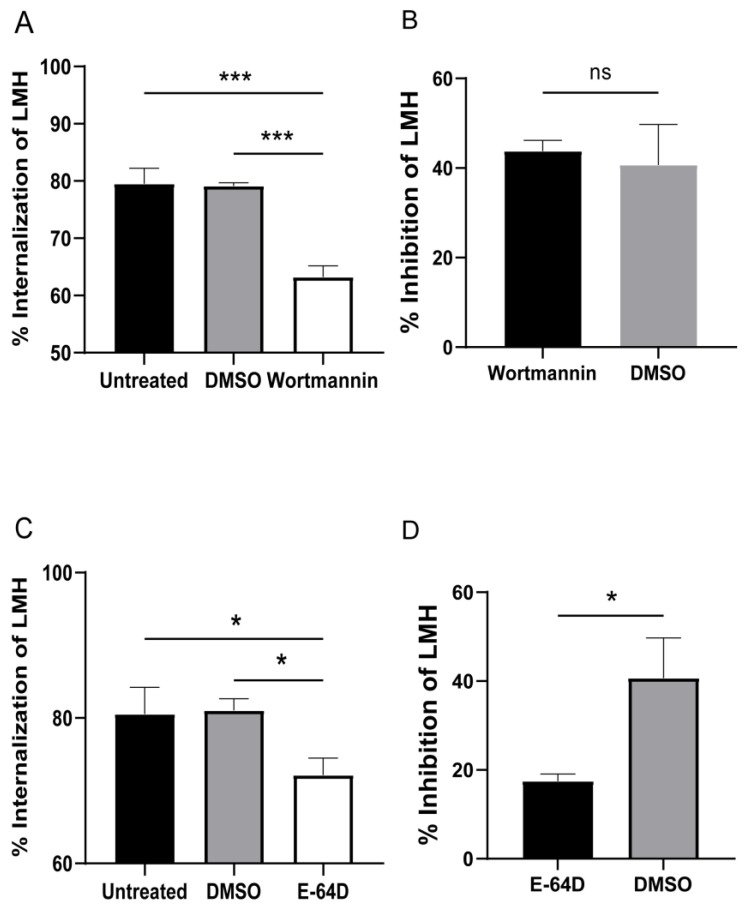
The effect of inhibitor treatment on the trogocytosis of *T. gallinae* was assessed. *T. gallinae* were pre-cultured with 30 nM wortmannin/30 nM E-64D or DMSO control for 1 h in panels (**A**,**C**), while panels (**B**,**D**) were utilized to evaluate the safety of both inhibitors on cells. Panel (**A**) shows that wortmannin significantly inhibited host cell damage caused by *T. gallinae*, and panel (**B**) presents the assessment of wortmannin’s impact on cell safety. Similarly, panel (**C**) demonstrates that E-64D significantly reduced host cell damage by *T. gallinae*, and panel (**D**) was used for the cytotoxicity assessment of E-64D. All presented data represent the mean ± SD of three replicate wells obtained from three donors and three separate experiments. Statistical significance is denoted by * *p* < 0.05, and *** *p* < 0.001. ns: no significance.

## Data Availability

All the data generated during the current study are included in the manuscript.

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
