# Peer review of "Trichomonas gallinae Kills Host Cells Using Trogocytosis"

_pathogens, 2023, doi:10.3390/pathogens12081008_

Round 1
Reviewer 1 Report
See the attached file with the comments.

It would be very useful if it can be polished by an English expert.
Author Response
Dear Reviewer:
Please see the attachment.

Reviewer 2 Report
The manuscript entitled "Trichomonas gallinae Kill Host Cells Using Trogocytosis" Title, abstract and overall rationale of work is written satisfactory. There are major concerns, which needs to be addressed before publication.
1) The abstract part is not satisfactory and author need to rewrite the abstract part and also need to write clear way.
2) Some important keywords are missing and author need to incorporate such as P13K inhibitor and others
3) In the introduction section: Author must be write italic in all species name in the whole manuscript see the line number 26.
4) Line number 36-38 author explained about the previously and they said other research group only focused on epidemiology, diagnosis and control and no one focus on mechanism of pathogen. It is very good if previous group/researcher focus on these things. Author need to revise the whole sentences and explain what he did in this manuscript.
5) Introduction section is not written satisfactory and author need to revise whole introduction part.
6) In the material methods section: Author need to correct the cell number (1×105 cells/mL) what is this?
7) In material method section author need to more elaborate especially flow cytometry part which kind of bead and dye they are used. Moreover, they also need to explain how many times this experiment repeated and they used triplicate well or duplicate wells.
8) What is CMFDA please write full name.
9) In the results section figure 1A showed three different channels but I don’t understand clearly what author want to explain. In this section author need to explain only results section instead of material and methods part.
10) Figure 1 author also need to add p-value
11) Result and Discussion section: This section author need to improve because author written the results part but they do not discuss properly and I saw the lack of discussion in this manuscript. I recommend author, they should elaborate the discussion part and author need to revise and compare the study with relevant study.
12) Add conclusion section and also write the future prospective.
13) There are too much problem in punctuation and typographical errors throughout in the manuscript. kindly correct it.
14) Most of the references are too old and author need to revise for example reference no 10, 42, and other. I suggest author to revise if other latest manuscript is available in the same information.
Author need to improve English quality.
Author Response
Dear Reviewer:
Please see the attachment.
Kind regards,
Cnen Xiang

Round 2
Reviewer 2 Report
The authors have addressed all the concerns raised in the previous version of the manuscript and the quality has much improved after incorporating required modifications. Therefore, the manuscript may be considered for publication in this Journal.
Now English is improved.
Author Response
Response to Reviewer 2 Comment
Point : Minor editing of English language required.
Response : We have corrected some errors in the article as you requested.